# Dismal pathological response to neoadjuvant chemotherapy in stage III breast cancer patients in Tanzania: A retrospective review

**Nashivai Kivuyo** [1]*, **Larry Akoko**[1], **John of God Mutajwaha**[1], **Caspar Haule**[2†], **Innocent Mosha**[3], **Mungeni Misidai**[1], **Daniel Kitua**[1], **Obadia Nyongole**[1], **Ally Mwanga**[1]

**1** Department of Surgery, Muhimbili University of Health and Allied Sciences, Dar es Salaam, Tanzania, **2** Department of Surgery, Muhimbili National Hospital, Dar es Salaam, Tanzania, **3** Department of Pathology, Muhimbili National Hospital, Dar-es-Salaam, Tanzania

† Deceased
* nashivaelias@gmail.com

## Abstract

Response to neoadjuvant treatment in breast cancer has been associated with good oncological outcomes. In Tanzania, a majority of breast cancer patients are diagnosed at stage III; hence, they almost always require neoadjuvant therapy. However, the response to neoadjuvant therapy in these patients remains unknown. This study examined the pathological responses in women with stage III breast cancer who underwent neoadjuvant therapy and identified sociodemographic and clinical predictors of the pathological response in this cohort. This hospital-based retrospective cohort study was conducted between December 2021 and April 2022. It included women with breast cancer who received neoadjuvant therapy and underwent surgery for breast cancer at Muhimbili National Hospital in Tanzania, from January 2018 through December 2021. Data analysis was performed using SPSS version 25. A complete pathological response was identified upon pathological review of the mastectomy specimen. Chi-square tests and Fischer's exact tests were used to evaluate the factors associated with a complete pathological response, with a p value of less than 0.05 indicating statistical significance. Ethical approval was obtained from the Muhimbili University of Health and Allied Sciences Institutional Review Board. The study complied with the Helsinki Declaration on studies involving human subjects. A total of 181 breast cancer patients were recruited for the study, with a mean age of 51±12.6 (28–89) years. A complete pathological response to neoadjuvant therapy was observed in 40 (22.1%) patients which is relatively lower compared to studies from Western countries. Disease stage at diagnosis was associated with response to neoadjuvant therapy, with those at stage IIIA showing better complete response than those at stages IIIB and IIIC indicating a need to improve diagnostic strategies to capture patients in the earlier stages to improve outcomes.

## Introduction

Breast cancer (BC) has emerged as a leading cancer globally, accounting for 11.7% of all new cancer diagnoses in 2020 [1]. While the incidence in Africa is relatively lower compared to

**Data availability statement:** Data cannot be shared publicly because of National IRB requirements. Data are available from the Muhimbili University of Health and Allied Sciences Institutional Data Access / Ethics Committee. For researchers who meet the criteria for access to confidential data, please contact the Directorate of Research, Publication and Innovation at Muhimbili University of Health and Allied Sciences using the email drpi@muhas.ac.tz or phone +255 222150302

**Funding:** The author(s) received no specific funding for this work.

**Competing interests:** The authors have declared that no competing interests exist.

that in high-income regions, mortality rates remain disproportionately high, particularly in sub-Saharan Africa (SSA) [2]. In Tanzania, BC is the second most common cancer after cervical cancer [3], with projected incidence and mortality rates expected to increase by 120% in the next two decades [4].

A significant proportion of BC patients in SSA present with stage III disease [5]. For this patient population, neoadjuvant therapy (NAT) followed by surgery and, when indicated, radiotherapy is the current standard of care [6]. NAT offers several advantages, including tumor size reduction, potentially enabling breast-conserving surgery (BCS) in cases where mastectomy would otherwise be necessary.

Recently, achieving a pathological complete response (pCR) to NAT has become a significant clinical goal, as it is associated with improved long-term survival and reduced risk of recurrence [7]. Factors such as the biological subtype of BC, disease stage, and treatment regimen can influence the likelihood of achieving a pCR [8,9]. Conversely, residual disease after NAT has been linked to an increased risk of local recurrence and the need for additional locoregional or systemic therapies [10–12].

In Tanzania, access to chemotherapy can be influenced by socioeconomic factors and the availability of specific regimens within the healthcare system. At Muhimbili National Hospital (MNH), patients diagnosed with locoregional BC are referred to Ocean Road Cancer Institute (ORCI) for NAT. While Ocean Road Cancer Institute offers chemotherapy, radiotherapy, and immunotherapy, NAT currently consists of neoadjuvant chemotherapy (NACT) alone leaving immunotherapy, hormonal therapy and radiotherapy for adjuvant setting.

Following completion of the planned NAT regimen, patients are evaluated for surgical candidacy and referred back to MNH for surgery. Currently, the assessment of the patient's response to chemotherapy within the NAT setting in SSA is not systematically documented. This study aims to address this gap by evaluating the pathological response to NAT in BC patients treated at MNH. The findings of this study will contribute valuable insights into the clinical characteristics and treatment outcomes of BC in this population, ultimately guiding the development of more effective and personalized treatment strategies.

## Materials and methods

### Study design and setting

This was a retrospective cohort study conducted at MNH, located in Dar es Salaam, Tanzania. The hospital sends its patients in need of non-surgical cancer directed therapies to ORCI also in Dar es Salaam, both being public institutions. Patients were included if they had a histologically confirmed BC, received NAT, underwent a mastectomy and had a post mastectomy histopathology assessment report.

### Participant identification and recruitment

We searched the electronic medical records (EMSs) of the two hospitals to identify patients who had received NAT during the period under review and entered this information into the first spreadsheet. Next, we examined the operating theater logbooks at MNH to identify patients who underwent breast surgery, and the names and types of surgeries performed were recorded in a second spreadsheet. We matched the two spreadsheets to identify stage III BC patients who received NAT and underwent BC surgery. For these patients, we retrieved their post-surgery histopathology reports and slide blocks for review, and those with missing histopathology slide blocks were excluded. The case notes of patients on the final list were reviewed to gather sociodemographic and clinical characteristics and pCR to NAT.

## Variables

The predictor variables included age at diagnosis in years calculated from the date of birth to the date of NAT initiation. The clinical stage was taken as documented in the case note at the time of initiation of NAT. The type of NAT regimen given in combination and the number of cycles were determined from chemotherapy charts. The histological grade of the pre-NAT samples and the expression levels of estrogen receptor (ER), progesterone receptor (PR), and HER2 were assessed in completely stained sections obtained at the time of diagnosis, and patients were classified into four molecular subtypes: luminal A for HR+/HER2-; luminal B for HR+/HER2+, triple-negative breast cancer (TNBC) for ER-/PR-/HER2-; and her2 enriched for HR-/HER2+ BCs.

The outcome variable was pCR. Because most patients did not undergo breast imaging before NAT, partial pathological response (pPR) could not be assessed because of failure to ascertain the number and size of tumors before NAT, thus limiting our outcome to only pCR versus no pCR. pCR was defined as the total absence of invasive cancer cells in the breast tissue and lymph nodes (ypT0/Tis, N0) in the mastectomy specimen. Patients with residual ductal carcinoma in situ (DCIS) and those with no evidence of residual invasive disease were also included in this category. Since the pathological response had not been reported routinely, a histopathological slide review was necessary. Hence, all the post-surgery histopathological blocks were reviewed by a senior pathologist to ascertain the pathological response, and in cases of disagreement with the initial report, a consensus was reached in consultation with another senior pathologist.

## Data collection

Two data abstractors were used to extract study-related variables using a predefined study questionnaire. The two complete abstraction sheets were compared per patient, and where there was a disagreement, either NK or LA would repeat the abstraction to resolve the discrepancy. After the data was complete and verified, de-identification was done by removing patients' names and hospital registration numbers from the Excel sheets, and data was transferred to the Statistical Package for Social Scientists (SPSS) for analysis.

## Data analysis

With SPSS software version 25, all the data were treated as categorical and summarized as frequencies with proportions. The chi-square test was used to compare the difference in the distribution of the dependent variable between two independent variables. Fisher's exact test was considered for frequencies less than 5 in any cell or when more than 20% of all the cells in the contingency table had an expected frequency less than 5. A p-value of less than 0.05 was considered statistically significant.

## Ethics approval

The study protocol was reviewed and approved by the Institutional Review Board of the Muhimbili University of Health and Allied Sciences with IRB number MUHAS-REC-11-2021-889. Permission to conduct the study was obtained from the MNH and ORCI administrations. No direct patient identifiers were used during data analysis following the deidentification process. Informed consent was waived due to the retrospective nature of the study.

## Results

Fig 1 below shows the recruitment of the study participants. A total of 376 patients received NAT at the ORCI and MNH between January 2018 and December 2021. However, 157 of

these patients were excluded from the study because they had no records of breast surgery at MNH. All 219 patients who underwent breast surgery underwent a mastectomy plus axillary dissection. Among these patients, 38 patients were excluded from the study because their post-surgery histopathology slide blocks were not available for review, making it impossible to ascertain their pathological response status. This left us with 181 patients for the study.

## Clinicopathological characteristics of the study participants

In Table 1, we summarize the clinical characteristics of the study participants. The mean age at the time of diagnosis was 51±12.6 (28–89) years: 35 (19.7%) were under 40 years of age while leading age group was between 40–49 years at 61 (33.7%) followed by those between 50–59 year at 37 (20.4%). TNM stage IIIC was the most predominant stage, representing 103 (56.9%) patients, followed by stage IIIB in 75 (41.4%) patients. Luminal A and TNBC molecular sub-types were the predominant subtypes, followed by Luminal B. Adriamycin (A), cyclophospha-mide (C), and taxane (T) were the most common combinations of ACT, AC, and AT, with the majority having been on ACT by 131 (72.4). None of the patients received targeted therapy in the NAT setting. pCR was seen only in 40(22.1%) of the patients.

## Association between clinicodemographic variables and pCR post-NACT

In Table 2 below, we present the pathological response rates as either complete or not complete and its associations with age, clinical stage, molecular subtype, treatment regimen and number of chemotherapy cycles administered. Clinical stage III A had the highest pCR at 100% compared to lower rates observed in IIIB and IIIC, a finding that was significant with p=0.01. In terms of age, the highest percentage of pCR (33.3%) was observed in the 70–80-year age group, whereas the lowest percentage (16.1%) occurred in the 61–70-year age group; however, this difference was not statistically significant. Among the molecular subtypes, varying percentages of patients who achieved a pCR were noted, with TNBC showing the highest response rate at 27.3% while Luminal A had the lowest pCR at 17.2%. Finally, with respect to the NAT regimens, the highest percentage of pCR was observed among patients receiving the AC-T regimen (24.4%), whereas lower rates were observed with the AC (16.3%) and AT (14.3%) regimens. pCR rate were higher among patients who received 5 cycles, 44.4%, compared to the other cycles but failed to reach significant levels.

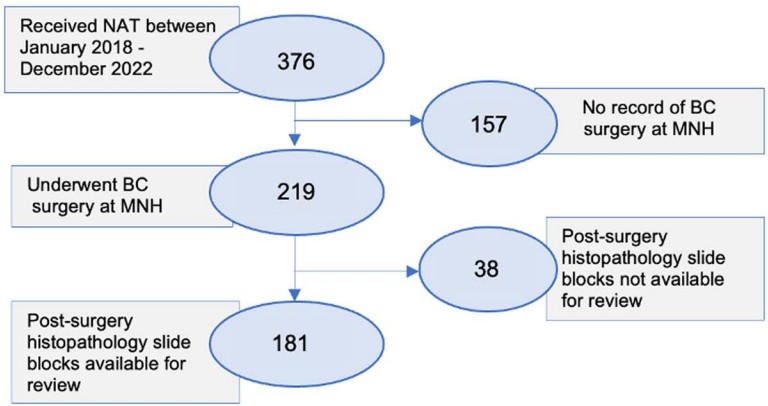

**Fig 1. Recruitment of study participants.**

**Table 1. Clinicopathological characteristics of the study participants.**

| Variables | Frequency (%) |
|---|---|
| **Age** | |
| <40 | 35 (19.7) |
| 40–49 | 61 (33.7) |
| 50–59 | 37 (20.4) |
| 60–69 | 31(17.1) |
| 70–79 | 12 (6.6) |
| >79 | 5 (2.8) |
| **Disease staging** | |
| IIIA | 3(1.7) |
| IIIB | 76 (42.0) |
| IIIC | 102 (56.4) |
| **Molecular subtype** | |
| Luminal A | 64 (35.4) |
| Luminal B | 33 (18.2) |
| Triple Negative | 66 (36.5) |
| HER2 enriched | 18 (9.9) |
| **NACT regimen** | |
| AC-T | 131 (72.4) |
| AC | 43 (23.8) |
| AT | 7 (3.9) |
| **Number of cycles** | |
| 4 | 35 (19.3) |
| 5 | 9 (5) |
| 6 | 24 (13.3) |
| 7 | 22 (12.2) |
| 8 | 91 (50.3) |
| **Pathological Response** | |
| pCR | 40 (22.1) |
| No pCR | 141 (77.9) |

Key A: Adriamycin; C: Cyclophosphamide; T: Taxane: NACT: Neoadjuvant Chemotherapy pCR: Pathological Complete Response

## Discussion

To the best of our knowledge, this is the first study in Tanzania that describes the response of BC to NAT and analyzes factors associated with pCR. The findings of this study can be used to infer the response of BC to NAT in the country since it was performed at the National Referral Hospitals, which receive patients from all regions of the country. While many studies report response following use of both chemotherapy and targeted therapy, the use of chemotherapy alone in this study makes it unique to allow comparison of response and drive changes in the management of BC in similar settings.

Only about 1 in 5 of our cohort attained a pCR to chemotherapy alone NAT. This rate of pCR is considered to be low compared to what is reported in literature of almost 50% [13–15]. Additionally, we argue that the pCR in our cohort may still be lower than the reported finding because we excluded 157 patients who did not undergo surgery at MNH post-NAT. Because all patients who receive NAT at ORCI are typically referred back to surgeons at MNH for

**Table 2.  Association between clinicodemographic characteristics and pathological response to Neoadjuvant Chemotherapy among post mastectomy patients at Muhimbili National Hospital, N= 181.**

| Variables | pCR (%) | No pCR (%) | p value |
|---|---|---|---|
| **Age** | | | |
| <40 | 9 (25.7) | 26 (74.3) | 0.253 |
| 40–49 | 13 (21.3) | 45 (78.7) | |
| 50–59 | 8 (21.3) | 32 (78.7) | |
| 60–69 | 5 (16.1) | 26 (83.9) | |
| 70–790 | 4 (33.3) | 8 (66.7) | |
| >79 | 1 (20.0) | 4 (80.00) | |
| **Disease stage** | | | |
| IIIA | 3 (100) | 0 (0.0) | 0.011 |
| IIIB | 19 (25.0) | 57 (75.0) | |
| IIIC | 18 (17.6) | 84 (82.4) | |
| **Molecular subtype** | | | |
| Luminal A | 11 (17.2) | 53 (82.8) | 0.394 |
| Luminal B | 7 (21.2) | 26 (78.8) | |
| Triple-Negative | 18 (27.3) | 48 (72.7) | |
| HER2+ | 4 (22.2) | 14 (77.8) | |
| **Regimen** | | | |
| AC-T | 32 (24.4) | 99 (75.6) | 0.453 |
| AC | 7 (16.3) | 36 (83.7) | |
| AT | 1 (14.3) | 6 (85.7) | |
| Number of cycles | | | |
| 4 | 7 (20) | 28 (80) | 0.226 |
| 5 | 4 (44.4) | 5 (55.6) | |
| 6 | 8 (33.3) | 16 (66.7) | |
| 7 | 3 (13.6) | 19 (84.6) | |
| 8 | 18 (19.8) | 73 (80.2) | |

**A: Adriamycin; C: Cyclophosphamide; T: Taxane**

surgical evaluation, we estimate that very few of these patients might have possibly undergone surgery at another facility. Hence, the possibility that these patients experienced disease progression to metastatic disease, which would eliminate the indication of surgical intervention, can be considered. Additionally, some of these patients might have abandoned NAT treatment due to various reasons including treatment toxicity or financial constraints. A prospective cohort study is warranted to complete understand treatment completion rates among patients undergoing NAT.

Several reasons can explain the low pCR rate observed in our study. One significant reason could be the relatively greater proportion of the luminal A subtype in our cohort. Our results revealed that the luminal A subtype had the lowest pCR at 15.6%, which aligns with other studies indicating that luminal A tumors respond less favorably to NAT than other subtypes. In contrast, TNBC patients presented the highest pCR rate, followed by HER2-positive enriched tumors and luminal B patients. Although the differences among these subtypes were not statistically significant, this finding is consistent with other studies showing that TNBC patients and HER2-positive patients generally respond better to NATs because of their aggressive biology [16].

Another explanation for the low pCR in our cohort is that none of the patients with HER2-positive BC received targeted therapy in the NAT setting. Given that studies have shown that the incorporation of targeted therapies such as trastuzumab can significantly improve pCR rates and overall survival outcomes [17], this omission is concerning. Potential barriers to this omission in LMICs may include limited access to medications, the high cost of targeted therapies, gaps in clinician awareness of updated treatment guidelines, and logistical challenges such as delays in obtaining HER2 status [18]. Investigating and addressing these challenges is crucial for improving pCR rates and optimizing treatment options for BC patients, ensuring that they receive the full spectrum of effective therapies available to improve their outcomes.

The study has demonstrated a notably younger population of patients with 1 in 5 being under 40-years of age among those that presented with locally advanced disease and were subjected to NAT. While this might be due to selection bias basing on disease stage, it echoes what has been reported in other countries in the regions [19] suggesting a genetic factor. This is In contrast to the picture in high income countries where BC is predominantly a disease of aging populations, with only 7% of cases diagnosed under 40 years, in spite of the intense screening programs [20]. Aggressiveness of disease in the younger age group and potential differences in tumor biology might explain the low response rates in our cohort given the relatively younger population it presents [21] However, age has not been documented in literature as a determinant of pCR [22], a finding that this study contradicts though with weak evidence. In contrast, other studies revealed that younger women with BC had higher overall survival rates, which was partly attributed to more favorable biological characteristics [23]. This divergence suggests that age-related factors may influence treatment outcomes differently in various populations, possibly due to differences in healthcare access, comorbidities, or biological responses.

While we included all stage III patients in our study, we observed very few patients with stage IIIA disease who received NAT and came back for surgery. This might be either due to the fact that some might have received an upfront surgery or a generally aggressive disease with such advanced presentation. When mastectomy is the only option offered, upfront surgery for stage IIIA is considered acceptable in the current guidelines [24]. However, clinicians must acknowledge the potential benefits of NAT even for stage IIIA BC patients., especially in the era of de-escalating breast cancer surgery and personalized care. The use of NAT can help reduce tumor size, thereby increasing the likelihood of achieving R0 resections and facilitating BCS [25,26].

Our study also revealed that complete response rates decreased significantly with increasing disease stage, with the majority of stage IIIC patients showing no response. Conversely, all patients with stage IIIA cancer who received NACT achieved pCR. Although this finding is statistically significant, it is important to note that the proportion of stage IIIA patients was very low, indicating a need for further research to assess outcomes on the basis of stage and response to NACT. Other regional studies and international studies also indicate that earlier-stage patients tend to respond better to NAT [27]. These findings emphasize the critical importance of implementing strategies to improve the early detection of BC to improve outcomes.

Most patients were offered a three-drug regime contain AC-T, which is the standard at ORCI with the rest being offered two drug regimens possibly due to limited accessibility, a common experience in LMICs facilities with out-of-pocket payments. As such, AC-T regime had the highest pCR rates compared to the other regimes offered. This finding is also in line with other studies, which showed that patients who received taxane-based regimens showed better response [22,28]. Ensuring adequate stocking to deliver AC-T based regimes in NAT

setting should be a priority given its better response. More studies are needed in this area in the local setting to optimize patients' outcomes

In our cohort, the type of breast surgery performed following NAT was mastectomy, even in those who achieved pCR. Ideally, patients with pCR post-NAT should be considered candidates for BCS. This contradicts the basis for NAT in the current setting where doing less extensive surgery is advocated [29]. This highlights the need for further investigation into the barriers preventing the utilization of BCSs in Tanzania. Potential barriers could include a lack of clinician awareness regarding BCS eligibility, prevailing cultural attitudes that favor mastectomy, and technical challenges, such as inadequate preoperative imaging and marking techniques, making it difficult for surgical teams to accurately locate and preserve healthy breast tissue during surgery [30]. Understanding patients preferences on whether or not to retain their breast is now becoming an important topic [31], one that has not been explored in our local setting. Addressing these barriers is essential for improving surgical options, outcomes, and quality of life for BC patients.

While this study provides valuable local data that can inform treatment strategies and improve patient management in Tanzania, it has several potential limitations. First, like all retrospective studies, there is a risk of selection bias and the inability to capture dynamic factors. Second, other factors that could influence the response to NAT, such as menopausal status, could not be evaluated because of insufficient documentation. Additionally, inadequate preoperative imaging prevented the reporting of pPR, which could have offered significant insights into the effectiveness of NAT by revealing the extent of the tumor response to treatment, even if a complete response was not achieved.

## Conclusions

This study provides valuable insights into the response to neoadjuvant chemotherapy in Tanzanian breast cancer patients. Key findings include a younger patient population compared to high-income countries, a relatively low pCR rate potentially influenced by tumor subtypes and limited access to targeted therapies, and a high rate of mastectomy even after pCR. These findings highlight the need for improved access to comprehensive treatment options, including targeted therapies, and the optimization of surgical approaches, such as promoting breast-conserving surgery where feasible. Further research is crucial to address the unique challenges faced by Tanzanian patients and improve breast cancer outcomes in this setting.

## Acknowledgments

Dr. Caspar Haule passed away before the submission of the final version of this manuscript. The corresponding author, accepts responsibility for the integrity and validity of the data collected and analyzed. We also acknowledge the faculty and residents of the Department of Surgery, Muhimbili University of Health and Allied Sciences.

## Author contributions

**Conceptualization:** Nashivai Kivuyo, Larry Akoko, John of God Mutajwaha, Caspar Haule, Mungeni Misidai, Daniel Kitua, Ally Mwanga.

**Data curation:** Nashivai Kivuyo, Larry Akoko, John of God Mutajwaha, Daniel Kitua.

**Formal analysis:** Nashivai Kivuyo, Larry Akoko, John of God Mutajwaha, Ally Mwanga.

**Methodology:** Nashivai Kivuyo, Caspar Haule, Innocent Mosha, Mungeni Misidai, Daniel Kitua, Obadia Nyongole, Ally Mwanga.

**Supervision:** Larry Akoko, Ally Mwanga.

**Writing – original draft:** Nashivai Kivuyo.

**Writing – review & editing:** Nashivai Kivuyo, Larry Akoko, Innocent Mosha, Mungeni Misidai, Daniel Kitua, Obadia Nyongole, Ally Mwanga.

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
