## [Decision Letter · Decision Letter 0]

18 Dec 2024

PONE-D-24-45078Dismal Pathological Response to Neoadjuvant Therapy in Stage III Breast Cancer Patients in Tanzania: A Retrospective ReviewPLOS ONE

Dear Dr. Kivuyo,

Thank you for submitting your manuscript to PLOS ONE. After careful consideration, we feel that it has merit but does not fully meet PLOS ONE’s publication criteria as it currently stands. Therefore, we invite you to submit a revised version of the manuscript that addresses the points raised during the review process.

We look forward to receiving your revised manuscript.

Kind regards,

Daniele Ugo Tari, M.D.

Academic Editor

PLOS ONE

Journal Requirements:

3. In this instance it seems there may be acceptable restrictions in place that prevent the public sharing of your minimal data. However, in line with our goal of ensuring long-term data availability to all interested researchers, PLOS’ Data Policy states that authors cannot be the sole named individuals responsible for ensuring data access (http://journals.plos.org/plosone/s/data-availability#loc-acceptable-data-sharing-methods).

4. Please amend your authorship list in your manuscript file to include author “Innocent Mosha”. 

5. We note you have included a table to which you do not refer in the text of your manuscript. Please ensure that you refer to Table 2 in your text; if accepted, production will need this reference to link the reader to the Table.

6. Please include a copy of Table 3 which you refer to in your text on page 6. 

Reviewers' comments:

Reviewer's Responses to Questions

**Comments to the Author**

1. Is the manuscript technically sound, and do the data support the conclusions?

Reviewer #1: No

Reviewer #2: Yes

2. Has the statistical analysis been performed appropriately and rigorously? 

Reviewer #1: N/A

Reviewer #2: Yes

3. Have the authors made all data underlying the findings in their manuscript fully available?

Reviewer #1: Yes

Reviewer #2: No

4. Is the manuscript presented in an intelligible fashion and written in standard English?

Reviewer #1: No

Reviewer #2: Yes

5. Review Comments to the Author

Reviewer #1: None of the patients received targeted therapy in the NAT setting. This is a retrospective study and without targetted treatments, we could not conclude the results. There is a selection bias in this cohort.

Reviewer #2: I am thankful for the opportunity to review the manuscript titled ‘Dismal Pathological Response to Neoadjuvant Therapy in Stage III Breast Cancer Patients in Tanzania: A Retrospective Review’. This is a well written manuscript from an LMIC, highlighting low PCR rates with NACT in stage 3 breast cancer patients undergoing modified radical mastectomy. The methods are appropriate for the study question at hand and the results are easy to follow. The authors demonstrate sound reasoning regarding potential reasons for low pCR rates in their cohort within the limitations of the study. I have few comments below.

ABSTRACT

Conclusion: States that ‘a relatively poor pathological response was observed’, however a comparison is not provided? Poor compared to what? Western counterparts? Please specify.

MAIN TEXT

Introduction

- Line 69: ‘Despite the relatively low incidence of BC in Africa, mortality remains high, and overall survival rates are disproportionally lower in SSA’ - A comparison is not provided again. Relatively low incidence compared to what? Also, would be better to specify the incidence and overall survival rates for clarity rather than providing a generic statement.

Methods

- Please define the duration of neoadjuvant therapy (number of cycles and lines)

Results

- Figure 2 is redundant and maybe safely omitted

- Was a modified radical mastectomy (mastectomy + axillary lymph node dissection) performed in all patients without an attempt at sentinel lymph node biopsy/targeted axillary dissection? This may reflect a difference in management from Western populations where there is an emphasis on de-escalation of complete axillary dissection.

- Do the authors have data regarding completion of planned chemotherapy cycles? This is an important covariate and should be compared between the two study groups.

- Do the authors have data regarding grade chemotherapy-related toxicities? While they document higher pCR rates with more aggressive neoadjuvant therapy (AC-T vs. AC or AT), it would be worthwhile to explore if this was at the expense of higher significant adverse rate. This may also be a reason for discontinuation of neoadjuvant treatment.

Discussion

- Can the authors comment about the utilization of screening mammography in Tanzania? What proportion of the cases were screening detected? In the absence of widespread screening, tumors are more likely to present at advances stages, which might be possible in their cohort

- Lines 257-268: In addition to recommending strategies for early detection, can the authors comment about measures to make modern neoadjuvant therapies available? For example, immunotherapy (Keynote 522 regimen) has been incorporated as the standard of care for neoadjuvant treatment of TNBC. Are there any measures which may be undertaken at a systems level to make these available in a LMIC setting?

6. PLOS authors have the option to publish the peer review history of their article (what does this mean? ). If published, this will include your full peer review and any attached files.

**Do you want your identity to be public for this peer review?** For information about this choice, including consent withdrawal, please see our Privacy Policy .

Reviewer #1: No

Reviewer #2: **Yes: ** Varun Bansal

---

## [Author Response · Author response to Decision Letter 1]

30 Jan 2025

We have adressed all the reviewer's comments. We have also modified the data sharing statement to include the third party contact who can be contacted to access data.

---

## [Decision Letter · Decision Letter 1]

3 Mar 2025

Dismal pathological response to neoadjuvant chemotherapy in stage III breast cancer patients in Tanzania: A retrospective review

PONE-D-24-45078R1

Dear Dr. Kivuyo,

We’re pleased to inform you that your manuscript has been judged scientifically suitable for publication and will be formally accepted for publication once it meets all outstanding technical requirements.

Kind regards,

Daniele Ugo Tari, M.D.

Academic Editor

PLOS ONE

Additional Editor Comments (optional):

Reviewers' comments:

Reviewer's Responses to Questions

**Comments to the Author**

1. If the authors have adequately addressed your comments raised in a previous round of review and you feel that this manuscript is now acceptable for publication, you may indicate that here to bypass the “Comments to the Author” section, enter your conflict of interest statement in the “Confidential to Editor” section, and submit your "Accept" recommendation.

Reviewer #1: All comments have been addressed

Reviewer #2: All comments have been addressed

2. Is the manuscript technically sound, and do the data support the conclusions?

Reviewer #1: Yes

Reviewer #2: Yes

3. Has the statistical analysis been performed appropriately and rigorously? 

Reviewer #1: Yes

Reviewer #2: Yes

4. Have the authors made all data underlying the findings in their manuscript fully available?

Reviewer #1: Yes

Reviewer #2: No

5. Is the manuscript presented in an intelligible fashion and written in standard English?

Reviewer #1: Yes

Reviewer #2: Yes

6. Review Comments to the Author

Reviewer #1: The early stage of diagnosis is associated with a complete pathological response, indicating a need to improve diagnostic strategies to capture patients in the early stages to improve outcomes. This manuscript was revised according to the reviewer suggestions.

Reviewer #2: I believe my comments have been addressed adequately. Thank you for your making the necessary modifications.

7. PLOS authors have the option to publish the peer review history of their article (what does this mean? ). If published, this will include your full peer review and any attached files.

**Do you want your identity to be public for this peer review?** For information about this choice, including consent withdrawal, please see our Privacy Policy .

Reviewer #1: No

Reviewer #2: **Yes: ** Varun V. Bansal

---

## [Editor Report · Acceptance letter]

PONE-D-24-45078R1

PLOS ONE

Dear Dr. Kivuyo,

I'm pleased to inform you that your manuscript has been deemed suitable for publication in PLOS ONE. Congratulations! Your manuscript is now being handed over to our production team.

Kind regards,

on behalf of

Dr. Daniele Ugo Tari

Academic Editor

PLOS ONE